# The Usage and Trustworthiness of Various Health Information Sources in the United Arab Emirates: An Online National Cross-Sectional Survey

**DOI:** 10.3390/healthcare11050663

**Published:** 2023-02-24

**Authors:** Mariam A. Almaazmi, Kamel A. Samara, Mohammed Jarai, Hussain Majeed, Hiba J. Barqawi

**Affiliations:** 1College of Medicine, University of Sharjah, Sharjah P.O. Box 27272, United Arab Emirates; 2Northeast Georgia Health System, Gainesville, GA 30501, USA; 3Department of Clinical Sciences, College of Medicine, University of Sharjah, Sharjah P.O. Box 27272, United Arab Emirates; 4Sharjah Institute for Medical Research, University of Sharjah, Sharjah P.O. Box 27272, United Arab Emirates

**Keywords:** health information sources, knowledge sources, United Arab Emirates, trustworthiness, Internet

## Abstract

**Background:** The increase in the quality and availability of health information as well as the accessibility of Internet-based sources, has driven growing demand for online health information. Information preferences are influenced by many factors, including information needs, intentions, trustworthiness, and socioeconomic variables. Hence, understanding the interplay of these factors helps stakeholders provide current and relevant health information sources to assist consumers in assessing their healthcare options and making informed medical decisions. **Aims:** To assess the different sources of health information sought by the UAE population and to investigate the level of trustworthiness of each source. **Methods:** The study adopted a descriptive online cross-sectional design. A self-administered questionnaire was used to collect data from UAE residents aged 18 years or above between July 2021 and September 2021. Health information sources, their trustworthiness, and health-oriented beliefs were explored through univariate, bivariate, and multivariate analysis in Python. **Results:** A total of 1083 responses were collected, out of which 683 (63%) were females. Doctors were the first source of health information (67.41%) before COVID-19, whereas websites were the first source (67.22%) during the pandemic. Other sources, such as pharmacists, social media, and friends and family, were not prioritized as primary sources. Overall, doctors had a high trustworthiness of 82.73%, followed by pharmacists with a high trustworthiness of 59.8%. The Internet had a partial trustworthiness of 58.4%. Social media and friends and family had a low trustworthiness of 32.78% and 23.73%, respectively. Age, marital status, occupation, and degree obtained were all significant predictors of Internet usage for health information. **Conclusions:** The population in the UAE commonly obtains health information from doctors who have been shown to have the highest trustworthiness; this is despite it not being the most common source used.

## 1. Introduction

The massive expansion of the Internet and social media, as well as its ease and wide accessibility, has led to a rise in health information-seeking behaviors. Despite a wide range of sources, accessing reliable health information remains challenging and elusive, with untrusted and uncredible sources potentially harming individuals’ health [1]. Therefore, researchers and clinicians aim to understand individuals’ health information patterns to better engage and promote successful behaviors [2]. Such behaviors include tackling health-threatening situations, making health-impacting decisions, and prioritizing preventive health habits.

Sources of health information can be categorized as Internet-based, entertainment-oriented, and information-oriented. Internet-based resources comprise broadly reaching mass data, such as blogs, websites, and social media, while entertainment-oriented include TV and podcasts [3]. Comparatively, information-oriented resources include healthcare providers or printed materials such as newspapers and brochures [4]. A global review study has shown that more than half of the public uses the Internet as a source of health information [5].

Health information research also includes evaluating a multitude of determinants for each resource. For example, the trustworthiness of a health information resource heavily determines its usage frequency and value, which in turn depends on various sociodemographic features [6]. Other research focuses on the different motives behind health information searching, which include symptom troubleshooting, searching before or after a clinical visit, or obtaining information for others [7,8]. For instance, individuals with long-standing diseases need to make decisions regarding their health; therefore, such patients tend to search more for information from multiple resources to make such decisions [3].

There is a paucity of research in the Gulf region on health information sources, with most focusing on the type of resource being used, with wide variation among the results being reported. A study conducted in Saudi Arabia showed that 87.6% of the participants relied specifically on doctors as their primary source of health information, whereas the Internet was not commonly used as a primary or secondary source [9]. However, a study targeting students in the Sultanate of Oman showed that the Internet and family members are more commonly utilized sources of health information compared to doctors and other experts [10]. These results align with those of a Kuwaiti university study that showed 92.6% of university students using the Internet as a health information source [11].

However, for the United Arab Emirates (UAE), there are no published results regarding primary or secondary general sources of health information. However, a study conducted by Figueiras regarding COVID-19 information resources exclusively found that only 20% would consult a physician [12]. Hence, understanding the different sources of health information used and the level of trust by the population in the UAE is necessary for helping individuals make informed medical decisions and evaluating healthcare options. Therefore, the aims of this study were to (a) evaluate the different information sources used by the population in the United Arab Emirates and their trustworthiness, (b) the impact of COVID-19 on the health information sources, and (c) explore the Internet as a health information source.

## 2. Methodology

### 2.1. Study Design and Target Population

A cross-sectional study was conducted between July 2021 and September 2021 to determine the sources of health information used by the population in the UAE. The eligibility criteria included (a) adults above the age of 18 years and (b) the ability to communicate in English and/or Arabic. Individuals younger than 18 years old and those who do not communicate in English or Arabic were excluded. This study and its protocols were reviewed and approved by the Research Ethics Committee of the University of Sharjah (REC-21-06-09-04S).

### 2.2. Questionnaire Development

A self-administered questionnaire was developed based on a review of the current literature on the topic [9,13,14,15,16,17]. It was developed in English and Arabic in Google Forms and was distributed online using different social media platforms. The questionnaire was initially developed in English, and translation was performed by two of the authors, who are fluent in both languages. It consisted of three sections, the first evaluating the demographic data and assessing their health status (presence of chronic diseases, frequency of health seeking, and the subjective rating of their health). It also made use of the well-established Single Item Literacy Screener (SILS). The second section investigated the different sources used before and after the COVID-19 pandemic, the frequency of usage, and the level of trust associated with each source. The third section evaluated the effect of those sources on the participants’ knowledge and health-related decision making. The questionnaire was pilot-tested several times, and provided feedback was assessed and incorporated where appropriate. To ensure the data had no missing variables, the questions were structured as “required” in Google Forms such that the participants could not move to the next question before answering the previous one.

### 2.3. Sampling and Data Collection

Sample size calculation is an essential part of any study to ensure adequate power. In this study, it was calculated using the well-established Cochran’s sample size formula, which is widely used, as can be seen in similar studies by the authors [18]. It states that for some standard normal variate $z_{1-\frac{\alpha }{2}}$ (calculated from the confidence interval), standard deviation SD, and error d, the sample size s can be calculated as follows:$$s=\frac{z_{1-\frac{\alpha }{2}}^{2}\times SD^{2}}{d^{2}}$$

Given the lack of any studies on the topic before, *SD* takes a value of 0.5 [19]. With a confidence level of 95% and a margin error of 5%, the estimated sample size in this study was calculated to be 385. This was increased to 440, assuming a 20% non-response bias. Given the non-probabilistic sampling technique used for recruitment, a total sample of 1000 was aimed for. The questionnaire was distributed through several online platforms such as e-mail, social media, and WhatsApp. A participant information sheet was presented before starting the questionnaire, and the agreement to fill out the questionnaire indicated consent to join the study. Additionally, no identifying data were collected to ensure participants’ anonymity.

### 2.4. Data Entry and Analysis

Data was exported from Google Forms to CSV format and processed in python using Matplotlib-v3.3.4, pandas-v1.2.4, and statsmodels-v0.12.2. Since all questions were required, there were no missing values. For univariate analysis, categorical variables were evaluated using percentages. Age was categorized into four groups, attempting to obtain meaningful groups (below 18; above 40) while ensuring that each group has a significant number of members to assist with statistical testing, as discussed later. Likert scale questions (ranked from 1 to 5), specifically the ones dealing with Internet frequency usage and health rating, were binned into three groups taking the middle score (3) as average. Hence, any score below the middle score was considered to be below average, while any score above was taken to be above average.

No outliers were detected. All demographic variables, health insurance status, health literacy, comorbidities, and health-oriented variables were used as predictors of Internet usage and knowledge source trustworthiness. Outcomes of interest were recoded into binary variables (frequency of Internet usage, doctor trustworthiness, social media trustworthiness, and Internet trustworthiness). This recoding was performed by combining the average and below-average groups into one and renaming it accordingly. This has the advantage of identifying factors associated with above-average trustworthiness and Internet frequency usage. Bivariate analyses were conducted to identify significant predictors using chi-squared tests. The significant predictors were then fed into a bivariate logistic regression model, which was evaluated using a log-likelihood ratio test. The cut-off for significance was a *p*-value less than 0.05.

## 3. Results

### 3.1. Demographics

A total of 1,083 responses were collected. A total of 63.07% of the participants were females, and 50.32% were between 19 and 29 years old. A third were UAE nationals, and nearly half were other Arab nationalities. Nearly 60% were residents of Sharjah and other northern emirates. A total of 39.06% of the respondents were students, and of those, 75.24% were students of health-related majors. As for occupation, 10.34% of all respondents were in the healthcare field. Of the total sample (1083), 72.85% have health insurance, and 84.30% have no long-term medical condition. Almost two-thirds had a normal reading ability of health literacy, which was assessed as the ease of understanding health information independently. More details regarding demographics can be found in Table 1.

### 3.2. Sources of Health Information

#### 3.2.1. Usage of Health Information Sources

When asked about their sources of health information before COVID-19, participants reported doctors as the most common source at 67.41%, followed closely by websites (62.51%) and social media (51.15%). As for websites and social media, examples such as World Health Organization, local government websites, and local electronic newspapers were used in the questionnaire to attempt to unify the perception of the participants about what is meant by websites is accurate. However, during the COVID-19 pandemic, websites and blogs became the most used source of health information, with 67.22% using it. Social media also increased to 63.99%, while doctors dropped to 59.19%. Figure 1 shows the frequencies for all health information sources surveyed.

The Internet was used mostly to learn about symptoms and diagnoses (79.22%), as well as to gain more information about COVID-19 (52.72%). Other uses of the Internet included subjects exploring treatment options (44.41%), gaining more information after a doctor’s visit (44.32%), researching self-treatment methods (37.12%), modifying health and lifestyle behaviors (28.53%), choosing a healthcare provider (27.89%), and deciding if a doctor visit is needed (26.87%). The main websites used were search engines (64.17%), international health agencies (48.29%), and local government websites (46.26%).

The determinants of Internet usage were explored through bivariate and multivariate analyses. Health orientation (*p < 0.0005*), occupation (*p < 0.0005*), marital status (*p = 0.00083*), health literacy (*p = 0.036*), long-term medical conditions (*p = 0.042*), place of residence (*p = 0.047*), and age (*p = 0.049*) were shown to be significant predictors and fed into a logistic regression model. All predictors except long-term medical conditions and health literacy remained significant. People who are older than 30 years (30–39 years; *p = 0.036*, *OR = 2.092 (95% CI: 1.051–4.162)* and >40 years, *p = 0.021, OR = 2.260 (95% CI: 1.131–4.513)*) were more likely to use the Internet more frequently. On the other hand, married (*p = 0.003, OR = 0.464 (95% CI: 0.280–0.769*)), non-healthcare (*p = 0.034, OR = 0.567 (95% CI: 0.335–0.960*)), students of other non-health-related majors (*p = 0.035, OR = 0.525 (95% CI: 0.289–0.957*)), and unemployed individuals (*p = 0.003, OR = 0.386 (95% CI: 0.206–0.725*)) were less likely to use the Internet. Results from the binary logistic regression model can be found in Table 2.

#### 3.2.2. Trustworthiness of Health Information Sources

Doctors were the most trustworthy source, with 82.73% stating them to be of high trustworthiness. Interestingly, while websites and blogs were the most common health information source, only 30.93% found them to be highly trustworthy. Overall, the least highly trustable health information source was social media, at 10%. Figure 2 shows the trustworthiness of the health information sources surveyed. With regard to doctors, social media, and the Internet, additional bivariate and multivariate analyses were conducted to explore the factors correlated with higher levels of trustworthiness.

With regards to doctor trustworthiness, health beliefs (*p < 0.0005*), marital status (*p < 0.0005*), health orientation (*p = 0.025*), occupation (*p = 0.010*), health consciousness (*p = 0.012*), long-term medical conditions (*p = 0.025*), and age (*p = 0.027*) were significant predictors at the bivariate level. Results of the multivariate regression showed that married individuals (*p = 0.009, OR = 0.450 (95% CI: 0.248–0.820*)) were less likely to trust doctors, while students of health-related majors (*p = 0.047, OR = 1.876 (95% CI: 1.007–3.494*)) were more likely to trust doctors, with all other variables being insignificant. Results from the binary logistic regression model can be found in Appendix A.

As for social media, age (*p < 0.0005*), nationality (*p < 0.0005*), sex *(p = 0.006),* occupation (*p = 0.006*), health beliefs (*p = 0.020*), and marital status (*p = 0.030*) all were found to be significant predictors of trustworthiness. However, at the multivariate level, health beliefs, marital status, and occupation were all shown to be insignificant. Hence, overall, individuals younger than 40 years of age (19–29 years; *p < 0.0005, OR = 0.161 (95% CI: 0.085–0.305)* and 30–39 years; *p = 0.026, OR = 0.333 (95% CI: 0.126–0.876*)), and non-Emirati Arabs (*p = 0.002, OR = 0.448 (95% CI: 0.267–0.749*)) were all less likely to trust social media. Results from the binary logistic regression model can be found in Appendix A.

Finally, the bivariate analysis showed occupation (*p < 0.0005*), sex (*p < 0.0005*), health literacy (*p = 0.003*), nationality (*p = 0.020*), and age (*p = 0.020*) to be significant in predicting Internet trustworthiness. When fed into the logistic regression model, only nationality was found to be insignificant. From the rest, only individuals with the normal reading ability of health literacy (*p = 0.018, OR = 1.443 (95% CI: 1.066–1.952)*) were more likely to trust the Internet. In contrast, individuals between 19 and 29 years of age (*p = 0.026, OR = 0.584 ((95% CI: 0.364–0.937)*), females (*p = 0.023, OR = 0.691 (95% CI: 0.503–0.950)*), students of non-health-related majors (*p = 0.002, OR = 0.385 (95% CI: 0.212–0.699)*), and unemployed individuals (*p = 0.033, OR = 0.496 (95% CI: 0.261–0.945)*) were less trusting of the Internet as a health information source. Results from the binary logistic regression model can be found in Appendix A.

## 4. Discussion

This study aimed to explore the different health information sources used by the population in the United Arab Emirates and to evaluate their trust in them. Before the COVID-19 pandemic, doctors were the most common source, followed closely by websites and social media. However, during the COVID-19 pandemic, the Internet rose to the first place and became the most common source of health information. However, doctors overall were still regarded as the most trustworthy source, with the Internet being considered partially trustworthy by the majority of participants. Age, sex, and occupation were all statistically significant predictors for the pattern of health information seeking and the perceived trustworthiness of each source.

There was a difference in the pattern of resource preference before and after the COVID-19 pandemic, which was also reported by another research conducted in the UAE during the pandemic. The researchers reported that while websites and social media platforms were the most used sources of health information, they were not the most trusted [12]. The increase in the use of the Internet as a source of information seeking during the pandemic could be explained due to the decreased accessibility to physically consult health workers and increased health anxiety [20]. This could also explain this study’s findings since the UAE did restrict access to non-emergency health services during the pandemic.

Although searching for more information regarding COVID-19 was a common reason behind Internet usage, it did not rank first in our findings. The most common reason was to learn about diseases’ symptoms and diagnoses. This presents a different picture compared with global studies where the Internet was mostly used complementarily after a doctor’s consultation [21]. Despite a fair percentage of participants (44%) supplementing their information from the Internet after a doctor’s visit, it was not the most common purpose of use; in fact, an equivalently large percentage (37%) reported searching for self-treatment methods over the Internet. When it comes to specific websites used over the Internet, the most used websites were search engines; interestingly, other studies in the Gulf region (Saudi Arabia and Qatar) reported similar results [13,22]. However, even while being the most used health information source over the Internet, both studies showed search engines to be not particularly well-trusted. While not explored in this study, trusted websites include those of personal doctors, medical universities, and federal medical organizations [16].

Overall, our results demonstrated that a vast majority of the participants (82.70%) regard doctors as the most credible source, whereas only a third of them ranked the Internet to be of high trustworthiness. This also matches a previous study where doctors ranked first in trustworthiness, followed by pharmacists [9]. Furthermore, our results showed that more than half the participants (56.23%) regard social media as partially trustworthy, in line with results from Saudi Arabia, where similar percentages distrusted the various social media sources [9]. Sbaffi and Rowley looked at the factors impacting the credibility of health information on social media platforms. Such factors included the authority of the author, the level of expertise in the field, and the objectivity of the posted information [6]. Finally, more than half of the participants partially trusted friends and family as a health information source, with another quarter reporting it being of low trustworthiness, making it the second least trustworthy source on the list.

Overall, trustworthiness and determinants of Internet usage are functions of multiple sociodemographic factors. One of the variables that influence the trust of individuals in specific health resources is age; older people tend to have less trust in any resources that are not healthcare providers [23]. Moreover, we found that older people are more likely to have less trust in social media as a health information source overall. In comparison, young people tend to prioritize readily available resources, probably due to their increased information needs, which cover social, physical, cognitive, and sexual self-development processes [24].

Preference for sources also differs among males and females as well; not only do females prefer consulting more than one source, but they also tend to search for information more than males. Studies conducted in Kuwait and Egypt showed a significant association between sex and utilizing the world wide web as a health information source, where females were more likely to seek health information compared to males [11,25]. As demonstrated by carpenter et al., one of the largest sex differences was that females tend to use medication package inserts as an information source more than males [15]. In this study, we found females to be less trustworthy of both the Internet and social media. Finally, level of education is another factor influencing health information behavior; the younger and the more educated an individual is, the keener they are to use diverse sources when searching for health information [22,26,27].

### Limitations

Every study has limitations that may affect the generalizability of the results; hence, a careful review of this study’s limitations follows. The participants may not be representative of the U.A.E.’s overall population due to the convenience sampling used. Moreover, no stratification was used to attempt to achieve specific percentages for the emirates, nationality, or occupations. For example, the proportion of the specific nationalities in the sample is not consistent with the actual proportion in the general population (where locals usually account for around 10% of the total population). However, care was taken during sampling to be inclusive and attempt to target all sectors of the community, and each group ended up having sufficient members for statistical analysis.

In addition, the sample consisted of a lower percentage of the older age groups, which may lead to bias. Given that older people may suffer from more long-term conditions and may need increased healthcare, this may affect the results and reveal different patterns of trustworthiness. Therefore, future studies could collect similar data from a larger sample and attempt to include older individuals. However, information access patterns by younger demographics are still relevant, given their unique healthcare challenges, as discussed above. No information was collected regarding the trustworthiness and frequencies of the websites being used by the participants. Similarly, no information regarding socioeconomic status (or an equivalent proxy) was collected. It is worth noting, however, that even then, the analysis above revealed several relations with the collected demographics. Finally, since this is a cross-sectional study, future prospective studies could be conducted to assess whether individuals consistently use the same sources of health information and the reasons behind it. Such studies can also evaluate other parameters of health information sources, such as accuracy and reliability. They can also attempt to address some of the limitations discussed here.

## 5. Conclusions

This research aimed to explore health information sources being utilized by the population in the UAE and the trustworthiness of each. While doctors used to be the most common health information source, the pandemic influenced health information-seeking patterns by prioritizing online sources such as social media and the Internet significantly increased. However, participants still recognized doctors as the most trusted source by the population in contrast to social media and friends and family, which were the least trusted sources. Finally, bivariate and multivariate analyses revealed a complicated interplay between source usage, source trustworthiness, and sociodemographic factors, most in line with global and regional studies.

## Figures and Tables

**Figure 1 healthcare-11-00663-f001:**
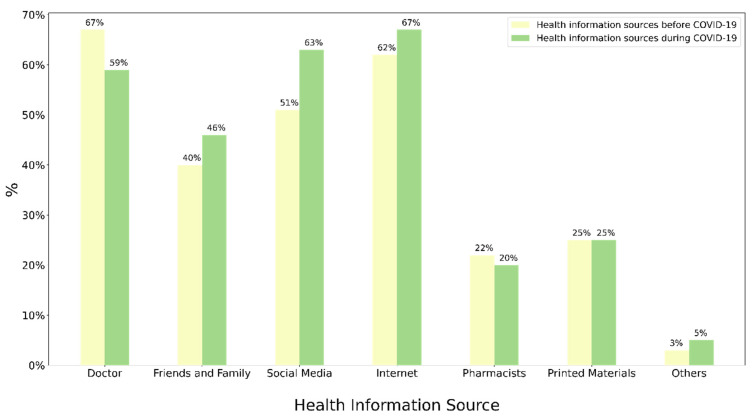
The distribution of health information resources usage before and after the COVID-19 pandemic.

**Figure 2 healthcare-11-00663-f002:**
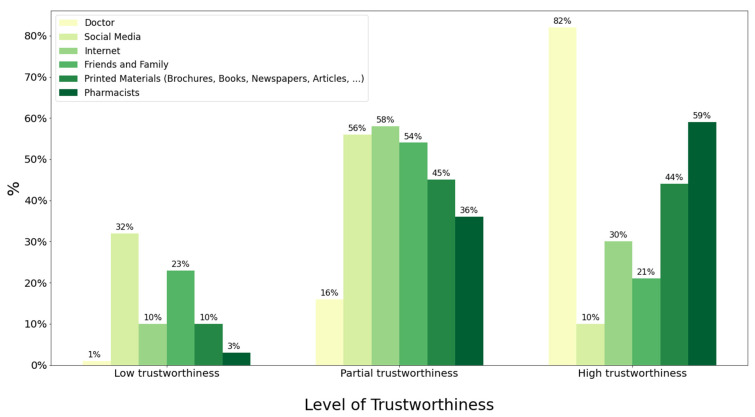
The level of trustworthiness for the different health information sources.

**Table 1 healthcare-11-00663-t001:** The various demographic variables of a total of 1083 responses with no missing values.

Feature	n (%)	Feature	n (%)
Sex		Field of Work	
Female	683 (63.07%)	Healthcare	112 (10.34%)
Male	400 (36.93%)	Non-healthcare	364 (33.61%)
**Highest Degree Obtained**		Student (health-related majors)	274 (25.3%)
High school or lower	410 (37.86%)	Student (non-health-related majors)	149 (13.76%)
Diploma/bachelor’s degree	574 (53.0%)	Housewife	98 (9.05%)
Postgraduate degree (MSc, Ph.D., etc.) or higher	99 (9.14%)	Unemployed	86 (7.94%)
**Age**		**Health Insurance**	
18 or below	124 (11.45%)	No	294 (27.15%)
19 to 29	545 (50.32%)	Yes	789 (72.85%)
30 to 39	159 (14.68%)	**Health Rating**	
40 or above	255 (23.55%)	Average or below	216 (19.94%)
**Marital Status**		Better than average	867 (80.06%)
Married	392 (36.2%)	**Health Literacy**	
Unmarried	691 (63.8%)	Limited reading ability	350 (32.32%)
**Nationality**		Normal reading ability	733 (67.68%)
UAE national	355 (32.78%)	**Long-term Medical Conditions**	
Other Arab	535 (49.4%)	No	913 (84.3%)
Non-Arab	193 (17.82%)	Yes	170 (15.7%)
**Place of Residence**		**Frequency of Internet Usage as a Health Information Source**	
Abu Dhabi	189 (17.45%)	Below average	107 (9.88%)
Dubai	261 (24.1%)	Average	238 (21.98%)
Sharjah and other northern emirates	633 (58.45%)	Above average	738 (68.14%)

**Table 2 healthcare-11-00663-t002:** The results of the logistic regression modeling the determinants of Internet use as a binary variable exploring its determinants. Rows with significant *p* values are bolded. OR: odds ration; CI: confidence interval; SE: standard error.

Determinants of Internet Use-Binary Logistic Regression (LR)						
Model Terms		$e^{\beta_{i}}/\mathrm{OR}$	95% CI for OR	SE	z-Statistic	*p*-Value
** $\mathrm{Intercept}\;(\beta_{0}$ ** **)**		**2.579**	**1.351–4.918**	**0.330**	**2.874**	**0.004**
**Long-term medical conditions**	No	-	-	-	-	-
	Yes	0.707	0.495–1.012	0.183	−1.896	0.058
**Age**	Younger than or equal to 18 years	-	-	-	-	-
	Between 19 and 29 years, inclusive	1.401	0.900–2.179	0.225	1.495	0.135
	**Between 30 and 39 years, inclusive**	**2.092**	**1.051–4.162**	**0.351**	**2.102**	**0.036**
	**40 years or older**	**2.260**	**1.131–4.513**	**0.353**	**2.308**	**0.021**
**Place of Residence**	Sharjah and other northern emirates	-	-	-	-	-
	Abu Dhabi	1.423	0.969–2.092	0.196	1.798	0.072
	Dubai	1.038	0.754–1.430	0.164	0.230	0.818
**Marital Status**	Unmarried	-	-	-	-	-
	**Married**	**0.464**	**0.280–0.769**	**0.257**	**−2.984**	**0.003**
**Health Literacy**	Limited Reading Ability	-	-	-	-	-
	Normal Reading Ability	1.101	0.828–1.464	0.146	0.658	0.510
**Occupation**	Healthcare (nurses, doctors, dentists, pharmacists, healthcare administration, etc.)	-	-	-	-	-
	Housewife	0.663	0.345–1.275	0.334	−1.232	0.218
	**Non-healthcare**	**0.567**	**0.335–0.960**	**0.268**	**−2.114**	**0.034**
	Student—health-related majors	0.882	0.509–1.528	0.281	−0.449	0.654
	**Student—other non-health-related majors**	**0.525**	**0.289–0.957**	**0.306**	**−2.104**	**0.035**
	**Unemployed**	**0.386**	**0.206–0.725**	**0.322**	**−2.959**	**0.003**
**Log-Likelihood: −655.80**	**Log-Likelihood of Null Model: −677.72**		**Log-Likelihood Ratio *p*-value: <0.0005**			

## Data Availability

All data related to this study are present in this paper.

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
