# Peer review of "The Usage and Trustworthiness of Various Health Information Sources in the United Arab Emirates: An Online National Cross-Sectional Survey"

_healthcare, 2023, doi:10.3390/healthcare11050663_

Round 1

Reviewer 1 Report

The paper is well written and the contents has relevance, mostly for the considered country. The problem is that the title is, in a certain way, misleading and we must be cautious about language used. The question would be; the study analyses the trustworthiness in the sense that information can/should be trusted or in terms of what people trust as source? (not that in fact it is a trustable/verified contents). That is the main problem. Also the study is not presented as was made. Since we don't know the questions, it is unlikely that we understand the point of the participating population. For instance, did the study considered why the participants have chosen such sources? was it because they were confined and had no access to other sources? was it because they were afraid to get exposed? The document doesn't clarify that. All those questions leave the study in low interest as we only know people had those choices and maybe it was not a question of trustworthiness but rather access, simplicity, fear, etc. On the other and, as mentioned before, we don't know if the sources are trustable in terms of reliability, we only know that people have chosen those sources.

Reviewer 2 Report

The usage and trustworthiness of various health information sources would be contributable research for realiable and accurate information for the improving information source. However, some concerns are followed as below ;

 1) In the part of Questionnaire development ;

It’s questionable how to assess the questionnaire. In manuscript, it’s divided into 3 sections and described each sections. Even though, it’s suggested that the process ofquestionnaire assessment to develope trustable questionnaire. Also, it’s recommended to include the full items of questionnaire in manuscripts as supplementary.

 2) Data entry and analysis ;

- In the final sentences, the last word ‘Results’ is need to confirm the necessary word

As questionnaire was developed in Google Forms and was distributed online using different social media platforms, it’s somewhat curious there’s no missing values. It might be some missing or unapproprate answer. Otherwise, is there any steps no going to next questions if no answer?

 3) Data entry and analysis

 a) As shown in Table 1, there’s item ‘Frequency of Internet usage as health information source’ and it’s categorize into 3 part as average. It’s wondering what average means. It has to be confirmed it is ‘Mean’ or other measure to categorize. Health Rating is the same. More detailed description is suggested.

 b) As shown in Table1, in Health Literacy item, Limited reading ability is 350 (32.32%). and in Long-term Medical Conditions items, No is 913 (84.3%). It’s like to influence variable. In Table2, Health Literacy item is included, but Long-term Medical Conditions is not included. It’s asking Long-term Medical Conditions is not included.

 c) For Figures;

-Since it is a bar graph, it would be better to have a space between the bars that the bar a little apart. Otherwise, it’s like histogram.

 d) For Tables of Binary logistic regression ;

As seen, it’s like binary multiple logistic regression.

- All notations has to be expressed again.

If taking an example, a) is Odds Ratio(OR)? , b) 95% CI is 95% C.I. for OR ? It’s suggested that re-expression of all notations and more expression or footnote of table using symbol if need

 e) As binary logistic regression was perfomed, it’s recommended to use odds ratio for the description of Table 2 in manuscript, not only just p-value.

 4) Discussion

It’s wondering if you haven't checked the accuracy of the health information source. In order to check the trustworthiness of the information source, the accuracy of the information source might be first checked, and then figuring out the relationship between the accuracy and trustworthiness of the information. For this, if it’s done in the research, more detailed description is required. If it’s not done. it’s required to listen the reason not to consider it. 

 Thank you.

Reviewer 3 Report

This is a well-structured paper.

Some suggestions:

In 2.1, 2.2, and 2.3 please give more details regarding the sample, can this sample be assumed representative to the people living in UAE and why. Other studies (as mentioned in introduction) examine these issues in specific group of people (ex. University/college). Did you consist a sample covering all the healthcare services users? As you describe in your results "84.30% have no long term medical condition" so in my opinion this sample is not representative to the healthcare services consumers or to the general population. In addition, 60% of the sample is high educated. 50% is on the age group of 19-29 and only 23,5% is over 40 (which is the part of the people who seek more healthcare services). I read the section 3.1 Validity (I would rename it as limitations) but you have to find a specific "label" to describe your sample because, as it is, confuses the reader that you had an representative sample of the UAE population. Title and Abstract must be revised.

Especially for 2.2, did you perform any reliability and validity tests for your questionnaire apart from the feedback assessment (lines 106-107)? If so, please provide some additional input. Based on your description of the questionnaire and the analyses that you present; this questionnaire includes some factors/dimensions. Please, give some information how you handled all these to ensure the reliability of the used measurement tool.

In 2.3 lines 109-110, how the required sample size was estimated. Based on which calculations? Additionally, please use references from similar studies performing the same minimum sample size estimation to explain. 

In 2.4, line 121: "Age was categorized into four groups." why? Which are these groups? Based on what did you perform the groups deviation.

Important: Please make sure that on the 2.4 you present an analysis (comparisons) that cover the aims of your study.

Results header is missing, I assumed that 2.5 is in the Results section.

I am not sure about the question regarding the comorbidities. Usually, we ask this question when we examine a specific diseases/condition, and we want to look if the presence of other diseases affect the outcome or health status. In this case, a question about chronic conditions will be fit much better.

In addition, I do not understand the "Health Literacy" -> "Limited Reading Ability"/"Normal Reading Ability" is this referred to the health info understanding?

Please rephrase the "Overall, the least trusted health information source was social media at 10%." this in not correct based on your results and the figure 2 presented. You can write something like: "Overall, the least HIGH-trustable health information source was social media at 10%." Please go through your manuscript to correct such mistakes.

About the categories sources, the "social media" and "internet" are not so clear. As you know on both of these sources you can find reliable and accurate content as well as personal opinions (which maybe biased). Please give more details on how the questions investigating these factors were structured and try to discuss the perception of the participants about the content. In other words, when the participants read these questions what did they had in mind? Did they understood the WHO webpage and ministry's of health twitter account or some content from unverified accounts/sources?

The categories in figure 1 and figure 2 are not the same. Internet / Websites and Blogs.

Finally, you can add some more limitations and after limitation you can present any future work.

Round 2

Reviewer 1 Report

Since questions were addressed, no further objections

Reviewer 2 Report

Thank you for all the efforts to revise the manuscripts according to the reviewer's opinion.  It would be the useful and contributable paper to improve the quality of health information area.